# Circadian Oscillations in Skin and Their Interconnection with the Cycle of Life

**DOI:** 10.3390/ijms24065635

**Published:** 2023-03-15

**Authors:** Andrew Salazar, Jörg von Hagen

**Affiliations:** 1Merck KGaA, Frankfurter Strasse 250, 64293 Darmstadt, Germany; 2Department of Life Science Engineering, University Applied Sciences, Wiesenstrasse 14, 35390 Gießen, Germany; 3ryon—GreenTech Accelerator Gernsheim GmbH, Mainzer Str. 41, 64579 Gernsheim, Germany

**Keywords:** circadian rhythms, entrainment, skin, bmal1, clock, chronobiology

## Abstract

Periodically oscillating biological processes, such as circadian rhythms, are carefully concerted events that are only beginning to be understood in the context of tissue pathology and organismal health, as well as the molecular mechanisms underlying these interactions. Recent reports indicate that light can independently entrain peripheral circadian clocks, challenging the currently prevalent hierarchical model. Despite the recent progress that has been made, a comprehensive overview of these periodic processes in skin is lacking in the literature. In this review, molecular circadian clock machinery and the factors that govern it have been highlighted. Circadian rhythm is closely linked to immunological processes and skin homeostasis, and its desynchrony can be linked to the perturbation of the skin. The interplay between circadian rhythm and annual, seasonal oscillations, as well as the impact of these periodic events on the skin, is described. Finally, the changes that occur in the skin over a lifespan are presented. This work encourages further research into the oscillating biological processes occurring in the skin and lays the foundation for future strategies to combat the adverse effects of desynchrony, which would likely have implications in other tissues influenced by periodic oscillatory processes.

## 1. Introduction

The term circadian rhythm, defined by Franz Halberg, a pioneer of chronobiology, in 1959, was originally adapted from Greek [1,2]. It is a hybrid of the words “circa” and “day”, meaning approximately 24 h or a day. In his later work, Halberg described biological cycles, which are an overlay of oscillations that occur in mammals. At the turn of the century, further work in the area led to the discovery that these oscillations took place under the control of molecular clocks. Initially, a hierarchical model was put forth where a master clock located in the brain was thought to dictate tact to the peripheral clocks in tissues and organs [3]. This hierarchical model has since been amended. The importance of the circadian rhythm was recognized in 2017, with the awarding of the Nobel Prize to Jeffrey C. Hall, Michael Rosbash, and Michael W. Young for their discoveries of molecular mechanisms controlling the circadian rhythm [4]. 

The findings of Halberg served as a foundation to understand aging as a multi-dimensional overlay of several clocks. Over time, dysregulation of these clocks occur, which is concomitant with a continuous and progressive loss of function of physiological and cellular processes. The rate of loss of function over time is influenced by intrinsic and extrinsic factors including nutrition and environmental factors [3,5,6,7,8,9]. In this review, we aim to summarize the research in the field of chronobiology, with a particular focus on skin biology. In addition to the circadian rhythm, this review also addresses the chronobiological changes that occur in the skin with the seasons over a given year, as well as the changes that occur over the lifespan of an individual.

## 2. Molecular Structure of the Circadian Clock

The mammalian circadian clock at a cellular level consists of at least 3 overlapping feedback loops (Figure 1) [3,8,10]. In the first loop, the core circadian clock proteins BMAL1 (basic helix–loop–helix ARNT llike 1; also called ARTNL, aryl hydrocarbon receptor nuclear translocator-like) dimerizes either with CLOCK (circadian locomoter output cycles kaput) or with NPAS2 (neuronal PAS domain protein 2) and then triggers the expression of PER (period circadian regulator; PER1-3), CRY (cryptochrome circadian regulator; CRY1-2), ROR (or RAR, related orphan receptor), NR1D1 (nuclear receptor subfamily 1 group D member 1; also called REV-ERE), DBP (D-box binding PAR Bzip transcription factor), and other clock controlled genes by binding to their E-box elements (5′-CACGTG-3′) in the promoter region. On reaching a critical concentration, the proteins PER and CRY dimerize to inhibit their own expression by preventing the binding of BMAL1:CLOCK to DNA, which results in an oscillation of these proteins [3,8,10,11].

In the second loop, the protein ROR binds to the RORE element 5′-(A/G)GGTCA-3′ in the promoter of BMAL1, CLOCK, and NFIL3 (nuclear factor, interleukin 3 regulated), resulting in their transcription. The binding of ROR to the RORE element is inhibited by NR1D1, and potentially also by related proteins from this family. This completes the second loop (Figure 1). The protein DBP, whose expression is under the control of BMAL1:CLOCK from the first loop binds to the D box (5′-TTATG(T/C)AA-3′) in the promoter region of PER. This binding is negatively regulated by NFIL3 from the second loop. Taken together, this is considered the third loop (Figure 1). Similar to PER1/2, CRY1 is regulated by a combinatorial mechanism involving both E-box and RORE, giving rise to a phase distinct from DBP and REV-ERB. The newly described DEC loop is ancillary to the core circadian loops, and is characterized by the expression of DEC and other circadian controlled genes, which are under the control of BMAL1:CLOCK. DEC, in turn, inhibits the binding of BMAL1:CLOCK to the E-box element, thereby regulating its own expression (Figure 1) [12,13]. Posttranslational modifications (PTMs) and proteasomal degradation of these core components of the molecular clock machinery are essential to maintain the oscillatory nature of these proteins and the proteins they regulate. These PTMs include phosphorylation, glycosylation, ubiquitination, acetylation, and SUMOylation, as reviewed by Hirano et al. [14]. In some cases, a further level of complexity is introduced in the form of crosstalk between these mechanisms; for instance, O-linked β-N-acetylglucosamine (O-GlcNAc) competing for the same serine and threonine residues as kinases for phosphorylation [14,15]. The PTMs that have been best characterized are those that are undergone by the PER proteins. One example of this is the phosphorylation of PER proteins by the casein kinases CKIδ and CKIε, along with CKIα, which has been identified from computational and high throughput screening campaigns [16,17,18,19,20,21]. Inhibiting these kinases leads to lengthening of the circadian period with an increase in the stability of the PER protein. Interestingly, the protein CK2 (also called CSNK2A2; casein kinase 2 alpha 2) has been shown to regulate the same protein in the PER accumulation phase, resulting in the shortening of the circadian period, as opposed to the phase in which PER levels decline where CKIδ and CKIε are implicated [17,18,19,20,21]. Further posttranscriptional regulators, targets, and their interactions continue to be elucidated; recently, HRD1 was suggested to be a regulator of BMAL1, although this protein is known to be regulated by CK2, PKCα, SIRT1, etc. [14,22]. A further mechanism for the regulation of the molecular circadian clock, which determines period length, is the ratio of CRY1 to CRY2 in cells, and the nuclear import rate of these CRY proteins as suggested by Li et al. [23].

The proteins of the circadian clock machinery continue to be characterized. This contributes to elucidation of further functions of the individual clock proteins, also contributing to a better understanding of the dynamics of their interactions and the context in which they exert influence. Recent studies have observed transcriptomic and proteomic oscillations, even with BMAL1 being knocked out [24]. However, this study is controversial, and contrasting results have been determined from the same dataset [25,26]. Nonetheless, circadian rhythms are complex, and are carefully orchestrated processes regulated at multiple levels with intricacies that continue to be unraveled. The investment of cellular resources and the multiple redundancies in the regulator mechanism indicate the importance of these processes with further compensatory mechanisms, and redundancies continue to be observed at the organism level [3,6,27,28].

## 3. Zeitgebers and the Circadian Clock in Mammalian Skin

The intracellular molecular clock oscillates in response to environmental signals known as ‘Zeitgebers’, derived from German and directly translating to ‘time giver’. As a result, in the study of circadian rhythm, time in days is often divided into ‘zeitgeber time’. Light is the primary zeitgeber. It is detected via the optical nerve, which then transmits signals of perceived light to the hypothalamic suprachiasmatic nucleus (SCN) [10]. This information is used to entrain the functional molecular clocks in peripheral tissues via the autonomic nervous system and the hypothalamus pituitary adrenal axis via hormones including glucocorticoids and catecholamines (epinephrine and norepinephrine) [29]. The hormones prolactin, growth hormone, and melatonin have been implicated in circadian signaling. Other zeitgebers that have been exploited in in vivo studies include sleep wake cycles, feeding and fasting regimes, and temperature (Figure 2). These zeitgebers are linked to the presence of light, as well as circulating levels of melatonin [30,31,32]. Ex vivo tissue retains circadian oscillation for days after it has been excised. This has been proven by studies using transcription analysis and bioluminescent clock gene reports [33,34,35,36,37,38]. To achieve synchrony of in vitro cells in culture, researchers have used serum shock, cAMP, and the glucocorticoid drug dexamethasone as zeitgebers (Figure 2).

Historically, it has been proposed that the clock hosted in the SCN is the master clock dictating the pace to the clocks present in the peripheral tissues. This hierarchical model has been reworked since it has been shown that light can entrain the circadian clock in these peripheral tissues even after SCN ablation or scarring, although oscillations were found to be lower in these cases [35,39,40]. Furthermore, the peripheral clocks can communicate with each other, achieving entrainment independent of the SCN. Recent research proposes a ‘memory’ and ‘response’ model where the peripheral clocks, particularly the clocks of the liver and skin, are capable of remembering the pace set by the SCN, but are also capable of adjusting their rhythm to stimuli perceived in the absence of a functioning SCN clock [39,40]. This research shows that light remains the primary zeitgeber, and since skin has an interface with light, its role as a mediator in circadian rhythm sensing is currently under scrutiny. The potential of UV, visible, and infra-red light to cause DNA damage, oxidative stress, lipid peroxidation, etc., in skin cells has been well characterized. Furthermore, UV light is also able to stimulate the production of melanin and melanocyte stimulating hormone (MSH) in the skin. This knowledge that skin cells have been known to interact with light makes it even more plausible that they can be directly entrained by light exposure. Photopigment neuropsin (OPN5), which is expressed in melanocytes and keratinocytes, has been proposed as a likely mediator of this interaction [35,41]. However, OPN5 would require 11-*cis* retinal for its function. In the retina, this is supplied by the retinal pigment epithelia from retinol via the enzyme retinoid isomerohydrolase (RPE65) [42]. The presence of this enzyme in mammalian skin is debated in the literature; thus, the source of 11-*cis* retinal in the skin is still unknown [43,44,45]. Further research is required to clarify the biochemistry and physiology of this intricate system.

## 4. Influence of the Circadian Clock on Immune Response of the Skin

The components of the circadian clock machinery are crucial to the development and functioning of a robust immune system. Indeed, most immune cell lineages have intrinsic clocks that govern their maturation, migration, differentiation, and function. An example is NFIL3, which is responsible for the development and maintenance of a population of interferon-gamma (IFN-γ) producing group 1 innate lymphoid cells and NK cells [46]. This, in turn, can be linked to the rhythmic activation of IFN-sensitive gene pathways in the skin, including a key transcription factor IFN regulatory factor 7 (Irf7) via Toll-like receptor 7 (TLR7). The induction of TLR7 is used in in vivo models to study psoriasiform inflammation [47,48]. Similarly, ROR-α expression is thought to be increased in activated phenotype T_reg_ cells in mouse skin [49]. Activated T_reg_ cells expressing ROR-α have been shown to attenuate the function of group 2 innate lymphoid cells that reside in the skin. This has been shown to limit allergic skin inflammation in models of atopic dermatitis mediated by type II cytokines including IL-4, IL-5, and IL-13, (Figure 3). Left unchecked, these interleukins would possibly involve the recruitment of T_H2_ cells into the skin, leading to increased CCL17 and CCL22 production [49,50,51]. Furthermore, the circadian clock determines the rhythm with which immune cells circulate or migrate into tissues. The adhesion molecules ICAM-1 and VCAM-1 on endothelial cells vary based on the degree of inflammation, as well as rhythmicity, and act as homing signals for leukocytes in homeostasis, as well as inflammation [52]. Moreover, in skin, CD44 appears to be the adhesion molecule that varies in a circadian manner and acts as a honing signal for leucocytes in endothelial cells that constitute the capillaries of the dermis [52,53]. This circadian rhythmic variation prevents overactivation of the immune system when an external challenge is unlikely, and preparation of the immune system in more active phases of the day when a host is more likely to be faced with a challenge of a pathogen [52]. This can also be seen in the enhanced expression of the antimicrobial peptide retinoic acid receptor responder 2 (Rarres2), cathelicidin antimicrobial peptide (Camp), and beta defensin 1 (Defb1) in phases of heightened activity [52,54]. Thus, although the immune system remains constantly vigilant and primed to mount a response to antigens, research into the influence of circadian rhythm on the immune system suggests an existence of a partition of the day into two phases. The first phase is one of heightened vigilance during waking hours where most activity occurs and an immune onslaught is most likely. This is followed by a recovery phase where resolution of inflammation and tissue repair occurs in the entire organism including the skin (Figure 3) [10].

The modulation of the two immunological pathways in skin has been shown to be influenced by glucocorticoids and nutrient intake, which act as zeitgebers. Of these, glucocorticoids have been studied in greater depth and have been linked to the central clock in the SCN [5,6,27,55]. The secretion of adrenocorticotropin (ATCH) from the anterior pituitary gland is under the control of the SCN [10]. This endocrine hormone, in turn, regulates the release of glucocorticoid hormones from the adrenal glands, which then give tact to the circadian clocks in peripheral tissues, as well as demonstrating immune regulatory function. However, ablation of the adrenal glands does not lead to loss of circadian oscillation in the skin and other peripheral tissue. In addition to the fact that skin can be directly entrained, this retention of circadian rhythm could be explained by the fact that keratinocytes of the epidermal layer in the skin are capable of regulating immune function by de novo synthesis of glucocorticoids via 11ß-hydroxylase (Cyp11b1), in addition to reactivation of inactive glucocorticoids via the enzyme 11ß-hydroxysteroid dehydrogenase type 1 (HSD11B1) [5,35].

**Figure 3 ijms-24-05635-f003:**
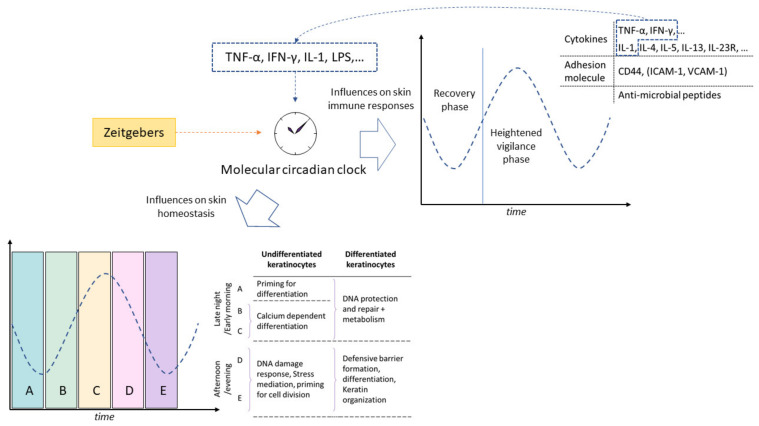
Influence of the circadian clock on skin immune response and homeostasis. The influence of the circadian clock on the immune responses in the skin allows for the day to be partitioned into a recovery phase and a heightened vigilance phase. Circadian oscillations of cytokines, adhesion molecules, and antimicrobial peptides have been observed. In the maintenance of epidermal homeostasis, the day can be divided into five succussive 4–5 h phases where distinct cellular processes occur. Further details can be found in [56]. Since the information in this review is a cumulation of information from studies performed in mice (nocturnal) models and cells, as well as human (diurnal) tissue and cells, the sinusoid in this figure serves a representational purpose only.

Thus far, the close interlinking of the immune responses and circadian rhythm has been established, and hence it can be correctly assumed that inflammation can disrupt the local, peripheral circadian clocks, as well as the central clock in the SCN. Recent reports implicate the NF-kB pathway in playing a central role in causing these perturbations [57,58,59]. In addition to this pathway, researchers have shown that TNF-α, IFN-γ, IL-1, and LPS are capable of disrupting the oscillations of a core clock gene and the genes that they control (Figure 3) [10,27,60,61,62,63,64,65,66]. In particular, TNF-α has been identified as a mediator of circadian phase changes. It alters the expression of a number of core components of the circadian clock machinery and has been attributed to the ability to inhibit the binding of BMAL1:CLOCK to its E-box promoter. Furthermore, disruption of the circadian clock can cause disruptions to the immune system. In a chronic jet lag mouse model, even a single exposure to jet lag was able to worsen the response to a high dose LPS challenge [65]. More specifically for skin, when tested in a mouse model for human allergic contact dermatitis, the T-cell mediated chronic hypersensitivity response was triggered by the disruption of the circadian clock. This pathology manifests together with heightened IgE level and increased mast cell numbers [67,68]. In another mouse model where psoriasis was induced via TLR7, CLOCK and Per2 were found to regulate the severity of psoriasis via the direct modulation of the expression of IL23R (Figure 3) [48]. In humans, epidemiological studies have also associated shift work, where the circadian clock is assumed to be disrupted, with higher risk of psoriasis [67,69]. These inflammatory reactions brought on by circadian disruptions are also likely to compromise the integrity of the skin, since, in human keratinocytes, TIMP3, which is a broad spectrum inhibitor of extracellular matrix (ECM)-degrading enzymes (MMPs, ADAM, ADAMTS), is likely to be under CLOCK control [3,6,27,47,70,71,72,73]. Thus, the magnitude of immune responses in the skin are profoundly impacted by the circadian system. In turn, disruption of the circadian rhythm of the skin leads to immune hyperactivity or an aberrant immune response that can manifest as pathologies such as dermatitis or psoriasis. A better understanding of the circadian rhythm and the immune system could help in the development of therapeutic approaches to treat diseased skin, especially since skin permeability also varies in a circadian manner [74]. Circadian transcriptome analysis has already delved into the oscillatory expression pattern of rhythmic genes in tissues (including skin) from a human, a non-human primate (baboon), and a mouse [75,76,77,78]. The core clock components and their immediate output targets were the most enriched transcripts across tissues. These studies also found that epidermal molecular oscillations are more robust than those of the dermal fibroblasts [79]. Non-sun-exposed skin showed the strongest nocturnal preference, whereas sun-exposed skin showed diurnal preference [76]. Furthermore, a majority of therapeutic targets are influenced by circadian oscillations, and many of the drugs that target these genes have a short half-life (<6 h), such that the circadian cycling of their targets could be consequential in the efficacy of administered treatment [75,76]. In an effort to take these oscillations into account, efforts have been made to monitor the circadian rhythm of a patient via biomarkers in the skin [79,80]. Although this approach has the potential to make big advancements in the field of circadian medicine, starting with the temporal adjustment of doses, the interference of circadian phases from peripheral tissue should be done cautiously.

## 5. Influence of the Circadian Clock on Skin Homeostasis and Stress Mediation

As in the case of the immune cells, circadian oscillations are observed in keratinocytes and melanocytes of the epidermis and the fibroblasts of the dermis [38,81]. The circadian clock machinery responsible for oscillations impact the metabolic processes of these cells and has an impact of tissue homeostasis. The epidermis is generated from epidermal stem cells in the basal layer that undergo asymmetric cell division, giving rise to either daughter stem cells or keratinocytes that will undergo a process of differentiation and desquamation to form the horny layer of the stratum corneum. It takes cells approximately 14 days for epidermal stem cells to end up as part of the stratum corneum. During this 2-week period, the process of differentiation does not occur continuously, but instead appears to occur in five sequential 24-h cyclic phases coordinated by the circadian clock. When studied via gene expression, each phase lasts for 4–5 h (Figure 3) [3,56,82]. For undifferentiated keratinocytes, the first phase includes cells being primed for differentiation with the upregulation of genes including *klf9* and *notch3* [56,83]. In the second and third phases, calcium dependent differentiation is triggered, together with the metabolic process associated with it. These three phases correspond to the late night to early morning hours, and vitamin D metabolism is also upregulated here. In the next two phases, genes associated with DNA damage protection and stress mediation are upregulated in undifferentiated keratinocytes, along with genes involved in preparing the onset of the next cycle of cell division and differentiation. In differentiated keratinocytes, the genes upregulated in the first three phases under circadian control remained similar to their undifferentiated counterparts, but included genes associated with DNA damage protection and repair, indicating constant vigilance against assault to the genetic code (Figure 3). In the next two phases, differentiated keratinocytes seem to shift their focus to building a defensive barrier with genes for differentiation and keratin organization being upregulated. This is likely to include the surface lipids of the skin that are under clock control and contribute to the skin barrier [56,84]. The differentiation process is not only dependent on the expression of the clock genes, but also their amplitude. The differentiation of the keratinocytes increases the amplitude of oscillation of PER1-2 and DBP, whereas that of BMAL1 is decreased. Disturbances to this oscillation, by overexpressing PER1 and PER2, or by decreasing the expression of CRY1 and CRY2, leads to spontaneous oscillations, which would perturb the division of the epidermal stem cells, and consequently also the homeostasis of the epidermis [35]. Autophagy is also closely linked to maintaining skin homeostasis. In the liver, the rhythmicity of autophagy is coordinated via C/EBPß, and in skin fibroblasts, the desynchrony of autophagy with age was reported by monitoring the gene expression of the marker LC3B and PER2 [70,85,86]. Although the interplay of autophagy, the circadian clock, and aging continue to be of interest, the role of these processes in skin cells has yet to be clarified [87].

Melanocytes and dermal fibroblasts have also shown to possess functioning circadian clock machinery, but the amplitude of oscillations appear smaller than that of keratinocytes. Despite this, the circadian clock plays a functional role in melanocytes by controlling the abundance of melanosomes, as well as the expression of melanin synthesis enzyme, *Tyrosinase* and the phosphorylation of MITF, which increases when BMAL1 or PER1 are silenced [88,89]. The protein OPN4 has been shown to affect the molecular clock components and their responsiveness to classical clock activators in melanocytes. Knocking out OPN4 in melanocytes resulted in rapid cell cycle progression and increased cellular proliferation, which correlated with the altered gene expression of MITF and the core circadian clock components [90]. The impact of the function of the circadian clock on dermal fibroblasts is yet to be characterized. However, the influence of circadian rhythm on the synthesis and secretion of Type I collagen, a major component of the ECM of the dermis, is already known [91,92]. Furthermore, the efficiency of migration and adhesion of fibroblasts modulated via actin dynamics was found to be circadian regulated [93]. This has been underlined by the correlation found from a database analysis indicating daytime wounds heal approximately 60% faster than wounds occurring at night. The nuclear protein NONO has been suggested as a possible molecular link between wound healing and the circadian rhythm of fibroblasts [94].

Thus, not only is normal function of skin impacted by circadian oscillations, but also the ability of skin to deal with stress is influenced by the cellular clock. As described above, the genes involved in DNA damage protection and repair in the epidermis are under clock control, as well as genes that mediate oxidative stress responses, in particular NRF2, the peroxiredoxins, glutathione peroxidase, and sestrins [95,96,97,98,99,100]. It can be hypothesized that these genes are under the control of the circadian clock so that the skin may prepare for stress mediation only when the onset of this stress is likely, i.e., during active hours with maintenance undertaken during non-active hours. This is corroborated with the fact that, in human skin, the activity of the DNA repair enzyme 8-oxoguanine DNA glycosylase (OGG1) was higher at night [101]. This is particularly important in the case of melanocytes, since they can accumulate DNA damage long after UV exposure, via melanin excitation [8,102]. In mouse skin, the xeroderma pigmentosum group A (XPA) protein, which is the rate limiting subunit of excision repair, exhibited circadian rhythmicity; thus, when mouse skin was exposed to UV irradiation, the likelihood of development of skin cancer after UV was linked to the time of day [103,104].

The evolutionary conserved hormone melatonin is also responsible for combating DNA damage and oxidative stress, as well as maintaining skin homeostasis [31]. This corelates with the finding that the circadian rhythm of melanin secretion is disrupted in psoriatic patients [105]. Since this molecule and its related metabolites are free radicle scavenges, it is capable of stress mediation [31,96]. Skin pigmentation and hair growth are also controlled by melatonin; its activity may also be influenced by the skin’s circadian clock [88,89,106]. Melatonin has also been implicated in the control of skin and body temperature in a circadian manner. In rat skin, the circadian clock machinery dermal fibroblasts is capable of using melatonin as an internal signal to fine-tune its oscillations, with temperature being used as an external queue [107]. Although melatonin is secreted from the pineal gland and synthesis is regulated by the degradation of arylalkylamine N-acetyltransferase (AANAT) in its synthetic pathway via light detected in the retina, the concentration of melanin in the skin can be far greater than that present in serum [108,109,110]. Therefore, further research is needed to disentangle the role of locally synthesized melatonin in the skin and its circadian clock.

After introducing the molecular mechanisms underlying the circadian rhythm, the following section will describe additional oscillations and lifespan associated processes, and how they interfere with skin relevant endpoints or molecular patterns.

## 6. Overlapping Oscillations and Underlying Individual Lifespan

### 6.1. Annual Clock

Physiological changes in human skin are a consequence of seasonality (Figure 4) triggered by several parameters such as light irradiation of various wavelengths with changing intensities over the year, temperature, temperature shifts between indoors and outdoors, humidity, and sweat resulting in, for example, a decrease in pH due to acidification and wind accelerated evaporation, thus modulating trans-epidermal water loss (TEWL) (Figure 4). The seasonal temperature changes modulate blood microcirculation and thus the accessibility to nutrients, which contributes to physiological skin changes (Figure 4). Recently, an effort was made to characterize these changes on the level of the transcriptome [76].

### 6.2. Challenges over Summer

An elevated temperature environment engages the skin thermoregulatory mechanisms. This includes increased activity of the sweat glands in the dermis that open out on the surface of the skin, leading to increased hydration levels, sebum dilution, and increased TEWL, which reduces the skin surface pH [111]. This could lead to skin itch [112], as the acidic pH of sweat results in an irritating effect on the skin by promoting Th2 and Th17-mediated inflammation and subsequent downregulation of filaggrin expression at the molecular level. The downregulation of filaggrin has a direct effect on moisture content in the skin since it is the precursor of the skin’s natural moisturizing factor (NMF) [113,114,115,116]. Interestingly, dry skin is mainly discussed as a result of the cold seasons, but not often considered over summer. In summer, the skin is typically exposed more intensely to sunlight, mainly UV-irradiation, which affects skin aging. During summer, the exposure to pollution is high as the climate and local weather conditions often do not allow effective air exchange and thus local pollution, also called urban stress, becomes more important, especially in crowded areas.

### 6.3. Challenges in Winter

The seasonal rhythm influenced by the duration of melatonin production in the summer is shortened in winter. Melatonin has been associated with hair growth, suppression of UV damage in skin cells, and wound healing [117]. Because it has antioxidant effects, topical melatonin has been used in wound healing, sun protection, and anti-aging products [117]. In winter, low relative humidity leads to high TEWL [118]. TEWL is often used as a parameter to assess the functional state of the epidermal barrier function. In vivo, it has been shown that exposure of skin to a low-humidity environment induces changes in the moisture content in the stratum corneum and skin surface pattern [119,120].

## 7. Intersection of Circadian Rhythms and Aging (Aging Clock)

The rhythmicity of multiple features are altered with age (Figure 5). This includes sleep, body temperature cycles, and locomotor activity [121,122]. For circadian rhythm, the amplitude of oscillation of clock controlled genes decreases with age in peripheral tissue, but not in the SCN, particularly when Per2 is being tracked [123]. This indicates that age-related circadian alteration in peripheral clocks are independent of the SCN clock. This is possible due to the weakened neurotransmission ability of the aged SCN neuronal network [123]. In aged skin, this problem is further compounded by the presence of senescent cells. Senescent cells show dampened circadian rhythmicity and are less efficient in the transmission of circadian signals to their clocks [124]. Therefore, senescence is implicated as the mechanism by which aging impairs entrainment of peripheral circadian clocks [124,125]. Furthermore, aged dermal fibroblasts secrete a unique aging-associated set of proteins, distinct from the canonical senescence-associated secretory phenotype. Among these include multiple candidates involved in inflammatory signaling and maintenance or alteration of the tissue microenvironment [126]. Thus, there is a high likelihood that aging via senescence is directly capable of disrupting the above mentioned immunological and stress mediatory pathways in skin. Since the rigidity of the tissue microenvironment is also impacted, aging also likely dampens the circadian clock of keratinocytes (which prefer a softer matrix) and fibroblasts (which prefer a more firm matrix) [127,128]. This being said, the role of the circadian clock in aging needs to be examined with very carefully designed animal studies, since core clock proteins have important roles in organismal development and maintenance. This would necessitate the use of conditional knockouts in place of conventional transgenic ‘null’ mice. For example, Bmal1 plays a crucial role in ocular and neural development [122,129], whereas CLOCK was found to play a role in heterochromatin stabilization, cell regeneration, and cartilage regeneration [130].

In the discussion of the circadian rhythm of aged skin, special considerations have to be made for skin stem cells (reviewed in [131]). The circadian oscillations of epidermal cells remain robust even under aged conditions, but they are rewired to adapt to the stressors associated with an aged environment, and remain committed to development and maintenance of the skin barrier through daily rhythmic cell division, in spite of DNA damage they many have incurred [132]. It has been hypothesized that maintenance of autophagic flux, which is under circadian control in liver cells, may help stem cells to adapt to stress in their environment and retain stemness [133,134]. However, this is yet to be proven in epidermal stem cells. In summary, the role of circadian rhythm and its relationship with skin aging is only now beginning to be understood. Hopefully, this review and the seminal research articles that it highlights foster further research in the area.

## 8. Conclusions and Outlook

Circadian rhythm and other biological periodic oscillations are a complex system of interconnected processes that are in fine balance. The disturbance of this continuously changing balance (intrinsic aging) leads to desynchronization events, that may be detrimental to phenotypic characteristics. A better understanding of the phases of these biological rhythms and the influence exerted by the relevant molecular players is important to develop agents that would be able to efficaciously resynchronize these rhythms. However, these potential modulations and their physiological consequences are yet to be thoroughly characterized.

Specifically, considering the circadian rhythm in skin that plays a role in stress mediation, any supplementation or support of the circadian-controlled cellular stress pathways would help rebalance the oscillations that occur in this tissue [95,96,97,98,99,100]. If the oscillation of the cellular circadian clock could be resynchronized, this would benefit the tissue not only in the mediation of stress, but also in the regulation of the immunological pathway, including the magnitude of their activation. This could prevent adverse effects of a desynchronized clock machinery, such as the atopic skin and disruption of the skin barrier that are observed in shift workers [8,69,135,136,137,138]. This knowledge would also help to recover asynchrony of skin cells after extrinsic stresses [139,140]. Nevertheless, additional research into the circadian rhythms of the skin and the players involved in the molecular signaling machinery is still necessary if aiming to achieve mitigation of desynchronization and its detrimental downstream effects.

Research on the entrainment of the peripheral circadian clocks, their interactions, and their impact on local tissue function, as well as on the central SCN clock, is still in its infancy. These interactions need to be better understood, not only to be able to prevent desynchronization of peripheral circadian clocks, but also to explore the possibility of resynchronizing the cellular circadian machinery via peripheral tissue, especially via the skin. These potential circadian modulations are gaining importance as we learn more about the important role of circadian rhythms in normal tissue functions, particularly relevant in modern society where we are persistently receiving potentially desynchronizing external stimuli over extended durations.

## Figures and Tables

**Figure 1 ijms-24-05635-f001:**
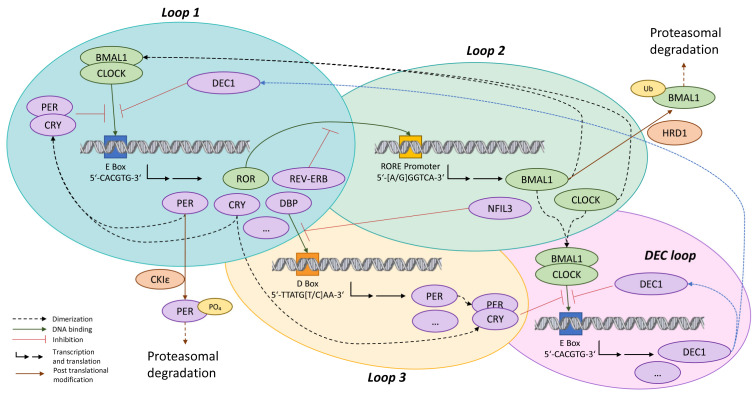
Molecular structure of the circadian clock. The circadian clock consists of three core feedback loops. The newly described feedback loop has also been depicted. **Loop 1:** BMAL1 dimerizes with CLOCK or NPAS2. These dimers bind to the promoter region E-box elements (5′-CACGTG-3′), triggering PER1-3, CRY, ROR, NR1D1, and DBP expression. PER and CRY dimerize on reaching a critical concentration, inhibiting their own expression, causing oscillation of the expression of these proteins. **Loop 2:** The transcription of BMAL1, CLOCK, and NFIL3 is triggered when ROR binds to the RORE element 5′-(A/G)GGTCA-3′ in their promoter region. This binding of ROR to RORE is inhibited by NR1D1. **Loop 3:** PER transcription is initiated when the D box element (5′-TTATG(T/C)AA-3′) in its promoter region is bound to by DBP (Loop1). This binding is negatively regulated by NFIL3 (Loop 2). **DEC loop:** The DEC protein that gives this loop its name, is responsible for its own oscillatory expression by inhibiting the binding of BMAL1:CLOCK to E-box elements (5′-CACGTG-3′) in its promoter.

**Figure 2 ijms-24-05635-f002:**
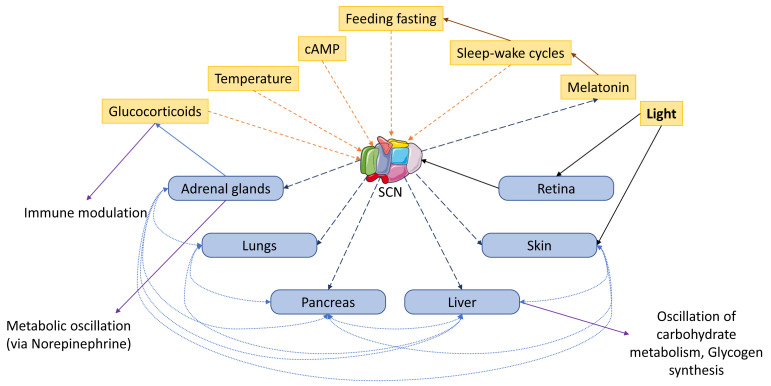
Zeitgebers and connections between central and peripheral circadian clocks. Light, the primary zeitgeber, gives tact to the central circadian clock in the hypothalamic suprachiasmatic nucleus (SCN) via the optical nerve. This, in turn, entrains the molecular clocks in peripheral tissues. The peripheral clocks can achieve entrainment independent of the SCN and can communicate with each other. Links between peripheral clocks examined in the literature are demarked with blue dashes. Sleep–wake cycle, feed and fasting, temperature, as well as melatonin, cAMP, and glucocorticoids can act as zeitgebers (illustration uses an element from Servier Medical Art: https://smart.servier.com/smart_image/suprachiasmatic-nucleus/ (accessed on 18 June 2021); licensed under a Creative Commons Attribution 3.0 Unported License).

**Figure 4 ijms-24-05635-f004:**
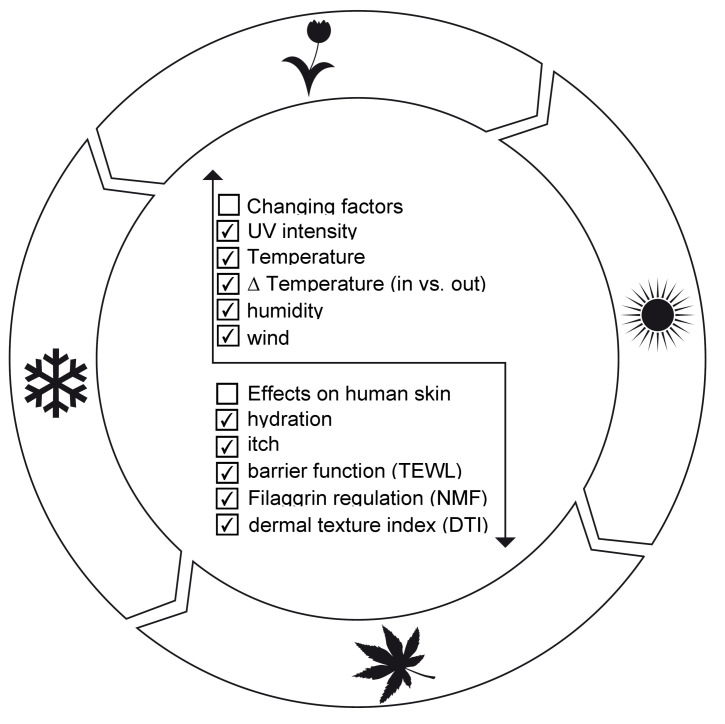
Seasonal variations or annual clock. During the annual four seasons, physical factors change, mainly based on the climate and weather situation, and individually directly affect the skin conditions (e.g., UV-intensity or wind). These results are based on exposure and its intensity to the listed effects on human skin, such as hydration.

**Figure 5 ijms-24-05635-f005:**
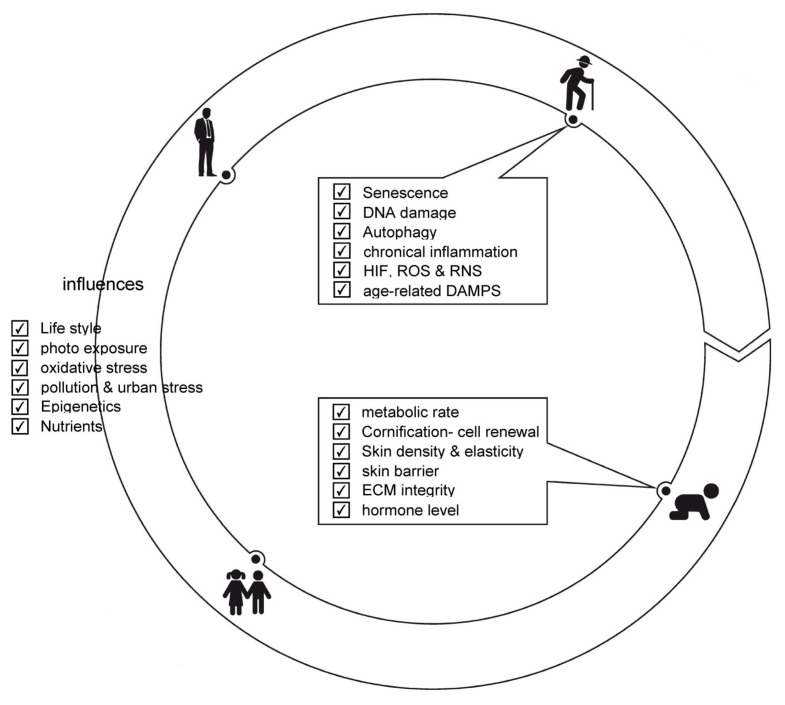
Aging clock. In infancy and early childhood, cellular processes are well balanced, and skin is in healthy state with given examples (metabolic rate to hormone level). The degree to which strong influences (e.g., from lifestyle to nutrients) affect the aging process of an individual are summarized. Cellular processes associated with aging range from, e.g., senescence to age-related damage-associated molecular patterns (DAMPS).

## Data Availability

Data sharing not applicable.

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
