# Peer review of "Circadian Oscillations in Skin and Their Interconnection with the Cycle of Life"

_ijms, 2023, doi:10.3390/ijms24065635_

Round 1

Reviewer 1 Report

1. Since most of the information is derived from results conducted in mice, it is inappropriate to specify “human skin” in the article title.

2. The abbreviations and corresponding full spelling in the text and table need to be carefully revised, especially for those of core clock genes (CLOCK, BMAL, PER, CRY, etc.). For example, CLOCK should be “circadian locomoter output cycles kaput”.

3. Skin sample has been used to predict individual circadian phase (Genome Med. 2020;12(1):73). Can the authors discuss the merit and demerit of skin circadian biomarker in assessing circadian rhythmicity as compare to other approaches?

4. Circadian transcriptome analysis has revealed the expression pattern of rhythmic genes in tissues (including skin) from human, baboon, and mouse (PLoS Biol. 2023;21(2):e3001986; Science. 2018;359(6381):eaao0318; Proc Natl Acad Sci U S A. 2014;111(45):16219-16224). By using these omic data, the authors are encouraged to present/discuss the oscillatory pattern of core clock genes and the enrichment of circadian pathways in the skins.

5. Page 6, lines 172 and 174, the word “honing” should be replaced with “homing”.

Author Response

Reviewer 1:

  1. Since most of the information is derived from results conducted in mice, it is inappropriate to specify “human skin” in the article title.

Response: The authors thank the reviewer for their feedback. The title of the manuscript has been modified accordingly.

  1. The abbreviations and corresponding full spelling in the text and table need to be carefully revised, especially for those of core clock genes (CLOCK, BMAL, PER, CRY, etc.). For example, CLOCK should be “circadian locomoter output cycles kaput”.

Response: The authors thank the reviewer for their feedback. These abbreviations have been checked and are now accurate.

  1. Skin sample has been used to predict individual circadian phase (Genome Med. 2020;12(1):73). Can the authors discuss the merit and demerit of skin circadian biomarker in assessing circadian rhythmicity as compare to other approaches?

Response: The authors would like to thank the Reviewer for their insightful comments. This study was included in the review and enriches the manuscript.

  1. Circadian transcriptome analysis has revealed the expression pattern of rhythmic genes in tissues (including skin) from human, baboon, and mouse (PLoS Biol. 2023;21(2):e3001986; Science. 2018;359(6381):eaao0318; Proc Natl Acad Sci U S A. 2014;111(45):16219-16224). By using these omic data, the authors are encouraged to present/discuss the oscillatory pattern of core clock genes and the enrichment of circadian pathways in the skins.

Response: The authors would like to thank the Reviewer for their insightful comments, which have greatly improved the article. A paragraph detailing the studies mentioned has been added and serves well to round off the manuscript.

  1. Page 6, lines 172 and 174, the word “honing” should be replaced with “homing”.

Response: The authors thank the reviewer for their feedback. This sentence have been revised for clarity.

Reviewer 2 Report

In this review manuscript, Salazar and von Hagen aimed to summarize studies on circadian oscillations in human skin, as well as their interconnections with the life cycle. They summarized the molecular circadian clockwork models, the circadian roles in immunological processes and skin homeostasis, the interactions between circadian rhythms, and annual and seasonal rhythms, and their effects on the skin. The authors also reviewed the aging clock These are intriguing topics. However, the current manuscript has numerous issues that must be carefully addressed.  

1. The title of the manuscript implies that circadian oscillations in human skin should be discussed. However, many circadian studies of human skin were not mentioned, such as circadian changes of skin metabolites, circadian regulation of DNA repairs, melatonin releases, and autophagy.

2. Numerous statements/descriptions should have appropriate references, such as “The term circadian rhythm defined by Franz Halberg, a pioneer of chronobiology, in 1959 was originally adapted from Greek.”

3. In Figure 1 and relevant text, it is hard to agree that DEC is the fourth feedback loop.

4. In Table 2, Upstream and downstream genes regulated by the circadian clock appear arbitrary; for instance, NAMPT is known to be regulated by CLOCK and BMAL1, and should be a downstream gene.

5. Line 52, “CLOCK (clock circadian regulator)”, what does clock circadian regulator mean?

6. Lines 102-103, “Recent studies have observed transcriptomic and proteomic oscillations even with BMAL1 being knocked out ”. As this study has been controversial, citing this paper without mentioning other papers/letters questioning the study, is one-sided and partial.

7. Lines 104-105, “Only 5-20% of genes expressed are said to be under circadian control (some of these”, Only 5-20% of rhythmically expressed genes are outdated and uncorrected.  

8. I suggest that the authors consult with a circadian biology expert and ensure that all the circadian statements/descriptions are correct.

9. Figures 1 and 2 should have detailed legends. Figures 4 and 5 should be in color and more informative.

10. The manuscript contains numerous awkward/erroneous words/sentences, such as, line 204, “The circadian control of the skin immune responses allow for the day to be partitioned…”. The manuscript should be carefully checked and ensure that English is correct.

Author Response

Reviewer 2:

In this review manuscript, Salazar and von Hagen aimed to summarize studies on circadian oscillations in human skin, as well as their interconnections with the life cycle. They summarized the molecular circadian clockwork models, the circadian roles in immunological processes and skin homeostasis, the interactions between circadian rhythms, and annual and seasonal rhythms, and their effects on the skin. The authors also reviewed the aging clock These are intriguing topics. However, the current manuscript has numerous issues that must be carefully addressed.  

Response: The authors would like to thank the Reviewer for their insightful comments, which have greatly improved the article.

  1. The title of the manuscript implies that circadian oscillations in human skin should be discussed. However, many circadian studies of human skin were not mentioned, such as circadian changes of skin metabolites: not much information to be found, circadian regulation of DNA repairs, melatonin releases and autophagy

Response: The authors would like to thank the Reviewer for their insightful comments, which have greatly improved the article. These details have been added and serves well to round off the manuscript.

  1. Numerous statements/descriptions should have appropriate references, such as “The term circadian rhythm defined by Franz Halberg, a pioneer of chronobiology, in 1959 was originally adapted from Greek.”

Response: The authors thank the reviewer for their feedback. The authors focused on publication in the last 10 years, in an effort to not needlessly inflate the citation list.

  1. In Figure 1 and relevant text, it is hard to agree that DEC is the fourth feedback loop.

Response: The authors thank the reviewer for their feedback. The authors did not intend to portray the DEC loop as a forth canonical feedback loop. These sentences have been revised for clarity.

  1. In Table 2, Upstream and downstream genes regulated by the circadian clock appear arbitrary; for instance, NAMPTis known to be regulated by CLOCK and BMAL1, and should be a downstream gene.

Response: The authors thank the reviewer for their feedback. Table 1 and 2 deleted to avoid any confusion

  1. Line 52, “CLOCK (clock circadian regulator)”, what does clock circadian regulator mean?

Response: The authors thank the reviewer for their feedback. These abbreviations have been checked and are now accurate.

  1. Lines 102-103, “Recent studies have observed transcriptomic and proteomic oscillations even with BMAL1 being knocked out ”. As this study has been controversial, citing this paper without mentioning other papers/letters questioning the study, is one-sided and partial.

Response: The authors thank the reviewer for their feedback. These sentences have been revised for clarity and opposing views have also been cited-

  1. Lines 104-105, “Only 5-20% of genes expressed are said to be under circadian control (some of these”, Only 5-20% of rhythmically expressed genes are outdated and uncorrected.

Response: The authors thank the reviewer for their feedback. These sentences have been revised for clarity 

  1. I suggest that the authors consult with a circadian biology expert and ensure that all the circadian statements/descriptions are correct.

Response: The authors thank the reviewer for their feedback. The entire manuscript has been check and revised where necessary.

  1. Figures 1 and 2 should have detailed legends. Figures 4 and 5 should be in color and more informative.

Response: The authors thank the reviewer for their feedback. Detailed Figure legends have been added to Figures 1 and 2.

  1. The manuscript contains numerous awkward/erroneous words/sentences, such as, line 204, “The circadian control of the skin immune responses allow for the day to be partitioned…”. The manuscript should be carefully checked and ensure that English is correct.

Response: The authors thank the reviewer for their feedback. These sentences have been revised for clarity.

Reviewer 3 Report

The manuscript of  Dr. Andrew Salazar, and Dr. Jörg von Hagen is opening on an audacious hypothesis that the skin is directly driven by light just like suprachiasmatic nuclei. The circadian clock molecular regulation are meticulously presented and well illustrated. The paper is well written and deserves publication in IMJS.
I am proposing suggestions of improvements to the authors.

Minor modifications

(1) « as well as rhythmical and act as honing signals for leucocytes in homeostasis as
well as inflammation [48]. Moreover, in skin, CD44 appears to be the adhesion molecule
that varies in a circadian manner and acts as a honing signal for leucocytes in endothelial
cells that make up the capillaries of the dermis [48, 49].
Do you mean « homing » instead of « honinhg » ?
(2) « and subsequent downregulation of filaggrin expressionat »
Please correct « expression_at »

Author Response

Reviewer 3:

The manuscript of  Dr. Andrew Salazar, and Dr. Jörg von Hagen is opening on an audacious hypothesis that the skin is directly driven by light just like suprachiasmatic nuclei. The circadian clock molecular regulation are meticulously presented and well illustrated. The paper is well written and deserves publication in IMJS.
I am proposing suggestions of improvements to the authors.

Response: The authors would like to thank the Reviewer for their insightful comments, which have greatly improved the article.

Minor modifications

(1) « as well as rhythmical and act as honing signals for leucocytes in homeostasis as
well as inflammation [48]. Moreover, in skin, CD44 appears to be the adhesion molecule
that varies in a circadian manner and acts as a honing signal for leucocytes in endothelial
cells that make up the capillaries of the dermis [48, 49].
Do you mean « homing » instead of « honinhg » ?
(2) « and subsequent downregulation of filaggrin expressionat »
Please correct « expression_at »

Response: The authors thank the reviewer for their feedback. These sentences have been revised for clarity.

Reviewer 4 Report

The manuscript entitled Circadian oscillations in human skin and their interconnection  with the cycle of life is presented for the peer review. Authors are right that a comprehensive overview of circadian periodic processes in skin is lacking in the literature. The paper is the effort to fill the gap. Circadian rhythm  is closely linked to immunological processes and skin homeostasis, and its desynchrony can be linked to perturbation of the skin. The interplay between circadian rhythm and annual, seasonal oscillations as well as the impact of these periodic events on skin is described here. The manuscript is well structured and clearly written . Good idea to mention new genes in circadian network, e.g. NFIL3 and HRD1. so, literature list is up to date. 

Some suggestions are dealing with light penetration to skin. could you add repair machinery in 2-3 phrases.

Also, I suggest OPN4 as one of key skin enzymes envolved in cell cycle progression , e.g. doi: 10.3390/cimb43030101.

Third, suggestion to add distortion of circadian rhythms in wound healing during light and dark cycles.

Minor checks circardian p.4 line 109

Author Response

Reviewer 4:

The manuscript entitled Circadian oscillations in human skin and their interconnection  with the cycle of life is presented for the peer review. Authors are right that a comprehensive overview of circadian periodic processes in skin is lacking in the literature. The paper is the effort to fill the gap. Circadian rhythm  is closely linked to immunological processes and skin homeostasis, and its desynchrony can be linked to perturbation of the skin. The interplay between circadian rhythm and annual, seasonal oscillations as well as the impact of these periodic events on skin is described here. The manuscript is well structured and clearly written . Good idea to mention new genes in circadian network, e.g. NFIL3 and HRD1. so, literature list is up to date. 

Response: The authors would like to thank the Reviewer for their insightful comments, which have greatly improved the article.

Some suggestions are dealing with light penetration to skin. could you add repair machinery in 2-3 phrases.

Response: The authors thank the reviewer for their feedback. A section of DNA damage via UV light and DNA repair mechanisms under the control of the circadian clock have been added

Also, I suggest OPN4 as one of key skin enzymes envolved in cell cycle progression , e.g. doi: 10.3390/cimb43030101.

Response: The authors thank the reviewer for their feedback. A paragraph on OPN4 has now been added.

Third, suggestion to add distortion of circadian rhythms in wound healing during light and dark cycles.

Response: The authors thank the reviewer for their feedback. A short section on wound healing in dark cycles exists in the manuscript. If any findings have been missed the authors would be happy to highlight them as well.

Minor checks circardian p.4 line 109

Response: The authors thank the reviewer for their feedback. This sentence have been revised for clarity.

Reviewer 5 Report

This is an interesting and timely review on a subject rarely addressed in the literature: the circadian organization of skin physiology. The message is clearly given and the hypotheses adequately discussed. A major lacking discussion is on melatonin direct activty on the skin. Such a discussion would help to complete the picture outlined.

Author Response

Reviewer 5:

This is an interesting and timely review on a subject rarely addressed in the literature: the circadian organization of skin physiology. The message is clearly given and the hypotheses adequately discussed. A major lacking discussion is on melatonin direct activty on the skin. Such a discussion would help to complete the picture outlined.

Response: The authors would like to thank the Reviewer for their insightful comments, which have greatly improved the article. A paragraph on melatonin has been added and serves well to round off the manuscript.